Spatial genetic structure in Pinus cembroides Zucc. at population and landscape levels in central and northern Mexico

García-Zubia Luis C. 1
Hernández-Velasco Javier 1
Hernández-Díaz José C. 1
Simental-Rodríguez Sergio L. 1
López-Sánchez Carlos A. 2
Quiñones-Pérez Carmen Z. 3
Carrillo-Parra Artemio 1
Wehenkel Christian wehenkel@ujed.mx 1
1 Instituto de Silvicultura e Industria de la Madera, Universidad Juárez del Estado de Durango , Durango , México
2 Department of Biology of Organisms and Systems, Mieres Polytechnic School, University of Oviedo, Campus Universitario de Mieres, C/Gonzalo Gutiérrez Quirós S/N , Mieres , Spain
3 Tecnológico Nacional de México, Campus Valle del Guadiana, Villa Montemorelos , Durango , México
Winkler Robert
Electronic publication date: 2019 Nov 6
Publication date: 2019
Volume: 7
Electronic Location ID: e8002
Received 2019 Jul 17; Accepted 2019 Oct 7
Copyright: ©2019 García-Zubia et al.
Copyright year: 2019
Copyright holder: García-Zubia et al.
License: This is an open access article distributed under the terms of the Creative Commons Attribution License, which permits unrestricted use, distribution, reproduction and adaptation in any medium and for any purpose provided that it is properly attributed. For attribution, the original author(s), title, publication source (PeerJ) and either DOI or URL of the article must be cited.
License URL: https://creativecommons.org/licenses/by/4.0/

Keywords: Gene flow, Seed stands, Sierra Madre Occidental, Isolation by distance

Funding: Comisión Nacional Forestal (CONAFOR), Mexico This work was supported by Comisión Nacional Forestal (CONAFOR), Mexico. The funders had no role in study design, data collection and analysis, decision to publish, or preparation of the manuscript.

==============================
Background

Spatial genetic structure (SGS) analysis is a powerful approach to quantifying gene flow between trees, thus clarifying the functional connectivity of trees at population and landscape scales. The findings of SGS analysis may be useful for conservation and management of natural populations and plantations. Pinus cembroides is a widely distributed tree species, covering an area of about 2.5 million hectares in Mexico. The aim of this study was to examine five natural seed stands of P. cembroides in the Sierra Madre Occidental to determine the SGS at population (within the seed stand) and landscape (among seed stands) levels in order to establish guidelines for the conservation and management of the species. We hypothesized that P. cembroides, in which the seeds are dispersed by birds and mammals, creates weaker SGS than species with wind-dispersed seeds.

Methods

DNA fingerprinting was performed using the amplified fragment length polymorphism (AFLP) technique. In order to estimate the SGS at population and landscape levels, we measured the geographical (spatial) distance as the Euclidean distance. We also estimated the genetic distances between individuals using the pairwise kinship coefficient.

Results

The results showed non-significant autocorrelation in four out of five seed stands studied (i.e., a mainly random distribution in the space of the genetic variants of P. cembroides at population level).

Discussion

SGS was detected at the landscape scale, supporting the theory of isolation by distance as a consequence of restricted pollen and seed dispersal. However, the SGS may also have been generated by our sampling strategy. We recommended establishing a close network of seed stands of P. cembroides to prevent greater loss of local genetic variants and alteration of SGS. We recommend seed stands of P. cembroides of a minimum width of 225 m.

Introduction

Gene flow mediated by exchange of pollen and seed within and among tree populations is necessary to maintain the long-term viability of forest species. Gene flow can stabilize local genetic variation and spread potentially adaptive genes (Van Dyck & Baguette, 2005). However, spatially limited gene flow can generate spatial genetic structure (SGS), i.e., the non-random geographical distribution of genetic variation caused by isolation by distance gene dispersal (Wright, 1938). Tree species with wind-dispersed seeds and that occur at low densities tend to generate stronger SGS in their seedlings than species with animal-dispersed seeds or that occur at high densities (Hamrick, Murawski & Nason, 1993). Moreover, limited seed dispersal that leads to neighbors being closely related also impacts demographic and reproductive processes (Hamrick, Murawski & Nason, 1993).

SGS analysis is a powerful approach to quantifying gene flow between trees (Segelbacher et al., 2010), thus clarifying the functional connectivity of trees at population and landscape scale (Van Dyck & Baguette, 2005). SGS can also be affected by mating systems, historical events and ecological and other evolutionary forces (such as selection and genetic drift), and it is therefore a crucial aspect of plant evolutionary processes and population dynamics (Epperson, 2003; Rousset, 2004; Vekemans & Hardy, 2004). The findings of SGS analysis may be useful for conservation and management of natural populations and plantations (McCue, Buckler & Holtsford, 1996).

The pattern of SGS revealed by autocorrelation analysis is greatly affected by the distance class width, sampling intensity and the total area sampled (Epperson & Li, 1996; Peakall, Ruibal & Lindenmayer, 2003; Double et al., 2005). Strong autocorrelation may occur when the spatial scale of sampling is smaller than the scale of the SGS (Cavers et al., 2005). In order to reduce bias due to the distribution, sampling along a fine mesh of transects in multiple dimensions and stratified sampling is recommended, thus combining high-density local sampling with a broader sampling coverage (Vekemans & Hardy, 2004). Overall, the line-transect scheme seems to perform slightly better than the simple-random scheme in parameter estimation and to be more efficient for encompassing broad spatial scales (Zeng et al., 2010).

SGS is also often associated with substantial stochastic variation due to genetic drift and a limited number of polymorphic genetic markers. Thus, gene markers such as allozymes, which provide few loci, are not adequate (Slatkin & Arter, 1991). Highly polymorphic markers (e.g., microsatellites) or markers facilitating many loci (e.g., amplified fragment length polymorphism [AFLP] markers) are therefore preferred in SGS analysis (Vekemans & Hardy, 2004). The advantage of using AFLP markers is that they are produced randomly throughout the whole genome and are highly reproducible and more sensitive than other markers for characterizing SGS (e.g.,  Leinemann et al., 2013; Leinemann et al., 2014). Moreover, AFLP markers are more precise (lower standard deviation) when the inbreeding coefficient is estimated independently (Mueller & Wolfenbarger, 1999; Hardy, 2003).

Pinus cembroides Zucc. (1832) is one of eleven to fourteen taxa known as pinyon pines and belonging to the subsection Cembroides, section Parrya, subgenus Ducampopinus (Gernandt, Liston & Piñero, 2003). The monoecious P. cembroides is widely distributed, covering an area of 2.5 million hectares in Mexico (CONAFOR, 2009). The largest populations are found in the states of Chihuahua, Coahuila, Durango, Nuevo León, Hidalgo and Zacatecas. Individual specimens of P. cembroides often grow in mildly acidic soils (mean pH 5.3, with H + representing on average 25% of the total exchangeable cations), under a warm xerophytic temperate climate (Wehenkel et al., 2015), almost always occupying transition zones between desert vegetation in arid climates and the more humid mountain forests (Rzedowski, 1978). The International Union for Conservation of Nature (IUCN) Red List guidelines indicate “minor concern” regarding the status of P. cembroides (IUCN, 2015). Pinus cembroides, which is also well-known for producing nutritious, cholesterol-free nuts (López-Mata, 2001; Luna-cavazos, Romero-Manzanares & García-Moya, 2008; Amr & Abeer, 2011), is suitable for reforesting arid, semi-arid and eroded areas.

Although phylogenetic analysis has been conducted (e.g.,  Flores-Rentería et al., 2013), to the best of our knowledge, only one study has investigated the SGS within and among P. cembroides stands (three) in the state of Durango, Mexico (Hernández-Velasco et al., 2017). These authors did not find any evidence of significant SGS at the local scale, suggesting that the genetic variants of these species are almost always randomly distributed in space, probably due to high wind pollination and seed dispersal by animals (Vekemans & Hardy, 2004). The significant level of SGS observed at a larger scale, however, may be the result of isolation by distance gene dispersal between populations.

The aim of this study was to examine five P. cembroides seed stands in the Sierra Madre Occidental, to determine the SGS at population (within each seed stand) and landscape (among seed stands) levels (Table 1) in order to establish guidelines for the conservation and management of this species. We hypothesized that P. cembroides, in which the seeds are usually dispersed by birds and mammals (Hubbard & McPherson, 1997; Richardson, 2000), creates weaker SGS than pine species with wind-dispersed seeds (Hamrick, Murawski & Nason, 1993).

Table 1 Location of the five seed stands in the Sierra Madre Occidental in the states of Chihuahua and Durango, Mexico.

Code	Forest stand	N	MD (m)	Stand size (ha)	Forest property, Municipality	Latitude (N)	Longitude (W)	Elevation (m)	
PC-MA	Mesa Azul	34	77	14.7	Bilaguchi, Guerrero	28°19′5.04″	107°39′43.57″	2,218	
PC-BQ	Baquiriachi	34	81	25.7	Baquiriachi, Balleza	26°57′0.20″	106°38′43.41″	2,369	
PC-MM	Mesa de la Majada	35	72	28.9	Ciénega Prieta, Guanaceví	26°15′8.60″	106°4′12.41″	2,631	
PC-CT	Cordón del Toro	35	54	8.6	El Toro y Anexos, Guanaceví	26°19′20.27″	105°5′3.14″	2,499	
PC-AD	Los Adobe	35	66	8.7	Garame de Abajo, Santiago Papasquiaro	24°58′22.79″	105°32′41.20″	2,331	
Notes.

N sample size

MD mean distance to the next tree

Data from Wehenkel et al. (2015).

Materials & Methods

Study area

The following five P. cembroides seed stands were sampled: Mesa Azul (PC-MA) and Baquiriachi (PC-BQ) in the state of Chihuahua, and Mesa de la Majada (PC-MM), Los Adobe (PC-AD) and Cordón del Toro (PC-CT) in the state of Durango (Table 1; Fig. 1). The seed stands were established according to the current Mexican norm for forest germplasm (NMX-AA-SC-169-2016; Hernández-Velasco et al., 2017). The five seed stands were growing on slightly acidic soil. The elevation ranges between 2,218 and 2,631 m above sea level in the study area, with annual rainfall between 498 and 514 mm. The mean temperature varies between about 11.0 and 13.1 °C (Table 2). The size of the seed stands ranged from 4.2 to 13.8 hectares. The mean distance to the next sampled tree varies from 54 to 81 m, depending on the stand considered (Table 1). Needles were collected from 34–35 randomly chosen trees in each stand. These sample trees, which were adult, dominant and occurred in the older age group, were a part of 130 previously selected adult phenotypes (trees) in each stand and were superior in terms of dimension, and pest-resistance, relative to the average tree in the same seed stands (for information about selection criteria of plus trees, see NMX-AA-SC-169-2016 and (Hernández-Velasco et al., 2017).

Figure 1 Location of the five Pinus cembroides seed stands under study.

Mesa Azul (PC-MA): black diamond; Baquiriachi (PC-BQ): grey square; Mesa de la Majada (PC-MM): small white diamond; Los Adobe (PC-AD): grey circle and Cordón del Toro (PC-CT): white triangle. Source: Available in the ARCGIS software (at Esri, DigitalGlobe, GeoEye, Earthstar Geographics, CNES/Airbus DS, USDA, USGS, AEX, Getmapping, Aerogrid, IGN, IGP, swisstopo, and the GIS User Community). Maps were created using ArcGIS® software by Esri. Copyright ©Esri. All rights reserved.

Extraction of DNA and genetic markers

DNA fingerprinting was performed using the amplified fragment length polymorphism (AFLP) technique, according to the protocol described by Vos et al. (1995) and modified by Ávila Flores et al. (2016). The DNA was extracted from the needles using the commercial DNeasy 96 plant kit (QIAGEN) and digested simultaneously with the restriction enzymes EcoRI and MseI. The primer combination E01/M03 (EcoRI-A/MseI-G) was used in the pre-AFLP amplification and the primer pair E35 (fluorescently-labelled with FAM) and M63+C (MseI-GAAC) in the selective amplification step.

All PCR reactions were conducted in a Peltier thermocycler (PTC−200 Version 4.0, MJ Research). The amplified restriction products were electrophoretically resolved in a genetic analyzer (ABI 3100) together with the GeneScan500 ROX (fluorescent dye ROX), and the size of the AFLP fragments was determined with GeneScan 3.7.1 and Genotyper 3.7. (Applied Biosystems). The AFLP bands were classified as present (1) or absent (0) in each individual, which was thus considered dominant or recessive (Simpson, 1997), i.e., detection of a band indicated the dominant genetic variant (the “plus phenotype”) (Krauss, 2000; Bonin et al., 2004).

Table 2 Some climate and soil conditions of the five Pinus cembroides seed stands studied in the Sierra Madre Occidental in the states of Chihuahua and Durango, Mexico (locations listed in Table 1).

Code	Mat (°C)	Map (mm)	Mmin (°C)	Mmax (°C)	Gsp (mm)	pH	
PC-MA	11.8	689	−5.1	28.2	510	5.54	
PC-BQ	12.3	627	−4.3	27.3	498	5.71	
PC-CT	11.8	635	−4.4	26.2	514	6.05	
PC-MM	11.0	644	−5.0	25.3	516	6.05	
PC-AD	13.1	661	−2.4	27.9	518	6.05	
Notes.

pH Degree of acidity

Mat Mean annual temperature

Map Mean annual precipitation

Mmin Mean minimum temperature in the coldest month

Mmax Mean maximum temperature in the warmest month and

Gsp Growing season precipitation, April to September

Analysis of the spatial structure of AFLP data

In order to estimate the SGS at population and landscape levels, the geographical (spatial) distance was measured as the Euclidean distance and the genetic distance was estimated from the pairwise kinship coefficient (Fij) (Hardy, 2003). Fij was measured using SPAGeDi v1.4 (Vekemans & Hardy, 2004).

For statistical reasons, we defined the width of spatial distance classes as including a minimum of 30 pairwise comparisons per distance class (Doligez & Joly, 1997). Thus, the smallest distance class width should not be less than 125 m at the local scale (i.e., within each of the five seed stands). The SGS statistics were therefore calculated for 125 m distance-class structures. However, as this autocorrelation technique are not capable of forecasting an appropriate analytical scale (Peakall, Ruibal & Lindenmayer, 2003; Double et al., 2005), we also calculated the SGS for distance class widths between 25 and 700 m (in 25 m steps from 25 to 300 m, in 50 m steps from 300 to 700 m), although the data did not comply with the condition of a minimum of 30 pairwise comparisons for the 25, 50, 75 and 100 m distance class widths.

At the large or landscape scale (among stands), the SGS was computed for distance class widths of six, seven, eight, 86 km and 168 km for a minimum of 2,900 tree pairs per distance class. For each spatial distance class, the 99% confidence interval (CI) was computed using 999 permutations (with SPAGeDi) (Manly, 2007). The probability value (P) was then computed for each spatial distance class and coefficient. After Bonferroni correction (Hochberg, 1988), the corrected critical p value (significance level α∗ = 0.0005) was calculated by dividing the original critical p value (0.05) by the number of comparisons or hypotheses (m = 100). Thus, the Bonferroni critical value of P was 1- α∗ = 0.9995 in this study (Hernández-Velasco et al., 2017).

The pairwise kinship coefficient (Fij) for dominant markers in diploids was computed and averaged over a set of distance classes to detect the SGS. The Fij coefficient, computed using SPAGeDi ver. 1.4, measures the occurrence of identical alleles at a given locus in a pair of individuals (Hardy, 2003), i.e., it estimates the ratio of differences of probabilities of identity (Rousset, 2002). If individuals are more closely related than individuals randomly chosen from the “reference” population, the relative kinship coefficients will have positive values. Consequently, negative values of the Fij indicate that i and j are less closely related on average than random individuals (Hardy, 2003).

PCoA analysis

The binary AFLP data matrix was also analyzed by Principal Coordinate Analysis (PCoA), and Nei’s Genetic Distance (Nei, 1972; Nei, 1978), which was determined using GenAlEx v6.503 (Peakall and Smouse, 2012), to compute the genetic separation between the seed stands and trees. The first two coordinates were used to graphically display genetic differentiation of populations and individuals.

Detecting AFLP loci under natural selection (outlier AFLP loci)

In the present study, the SGS of AFLP markers under natural selection may be an indicator of environmentally adapted provenances (Epperson, 1992; Stingemore & Krauss, 2013). Candidate AFLP loci under natural selection were determined using differences in allele frequencies between populations, and the multinomial Dirichlet model and the Reversible Jump Markov Chain Monte Carlo algorithm were implemented in BayeScan v2.1 (Foll & Gaggiotti, 2008). AFLP markers were used as purely dominant binary data, but the inbreeding coefficients (FIS), used to estimate allele frequencies, were not able to be estimated from binary data. Therefore, BayeScan allowed FIS to move freely within its prior range in order to still incorporate the uncertainty associated with this parameter.

A negative locus-specific component (A) value with posterior probability >0.99 indicates possible balancing or purifying selection, and positive values of the A and posterior probability >0.99 indicate diversifying selection (false discovery rates <0.01) (Foll & Gaggiotti, 2008; Foll et al., 2010). We used the parameter values of the chain and the model reported by Friedrich et al. (2018): output number of iterations (5,000), thinning interval size (10), pilot runs (20), length of pilot runs (5,000), additional burn in (50,000), prior odds for the neutral model (10), a lower boundary for uniform prior in the inbreeding coefficient FIS (0) and a higher boundary for uniform prior in FIS (1).

Clark and Evans aggregation index

The aggregation index (CE) proposed by Clark & Evans (1954) was calculated using Spatial Genetic Software v1.0. The values obtained indicate the spatial structure, where CE <1 represents an aggregated distribution, CE = 1 indicates a random structure and CE >1 indicates a regular distribution. The statistical significance of CE was calculated using permutation tests (Degen, 2000).

Results

The combination of the AFLP primers resulted in 281 polymorphic bands of 75 - 450 bp across the 173 individual Pinus cembroides trees analyzed. No candidate AFLP loci under differential selection (putative adaptive AFLP) were detected (posterior probability >0.99 and false discovery rates <0.01).

At the fine scale (within each of the five seed stands), significant autocorrelation (P >0.9995, after Bonferroni correction) was only observed in PC-MA. In particular, significant autocorrelation was only detected in the first five distance class widths (i.e., 0–100 m, 0- 125 m, 0–150 m, 0–175 m and 0–200 m of the class sizes 100, 125, 150, 175 and 200 m) and the distance classes 50–75 m and 50–100 m (Table 3). Fij was weakly positive in the first distance class in five distance class widths (0–75 m to 0–175 m) in PC-MA and in a few other distance classes in PC-CT and PC-AD. However, this index became more negative in larger distance class widths (>0–125 m in PC-AD and >0–200 m in PC-MA) (Tables 3 and 4). The values of the aggregation index of Clark and Evans (CE) indicated that the spatial distribution of the seed trees under study was only clumped in PC-MA and PC-AD (Tables 3 and 4).

Table 3 Analysis of the genetic spatial structure within the Pinus cembroides seed stands PC-MA, PC- BQ and PC-MM, considering the first class in distance class widths of 25–700 m, with 999 permutations.

The analysis was conducted using SPAGeDi v1.4 (Vekemans & Hardy, 2004).

Fijvs. spatial distance	
Distance(m)	PC-MA	PC-BQ	PC-MM	
	CE= 0.61***	CE= 1.28 **	CE= 1.05 ns	
	P(Fij) <CI	Fij	P(Fij) <CI	Fij	P(Fij) < CI	Fij	
0–25#	0.277	−0.085	0.523	−0.029	0.473	−0.031	
0–50##	0.262	−0.061	0.514	−0.029	0.482	−0.031	
0–75	0.999	0.041	0.393	−0.037	0.601	−0.025	
0–100	0.9999+	0.030	0.553	−0.027	0.608	−0.026	
0–125	0.9999+	0.020	0.244	−0.038	0.921	−0.016	
0–150	0.9999+	0.009	0.195	−0.038	0.867	−0.019	
0–175	0.9999+	0.006	0.276	−0.034	0.912	−0.020	
0–200	0.9999+	−0.001	0.498	−0.030	0.838	−0.023	
0–225	0.993	−0.007	0.520	−0.031	0.527	−0.029	
0–250	0.994	−0.011	0.755	−0.027	0.549	−0.029	
0–275	0.984	−0.015	0.448	−0.031	0.635	−0.028	
0–300	0.973	−0.018	0.532	−0.030	0.652	−0.028	
0–350	0.969	−0.019	0.695	−0.029	0.344	−0.031	
0–400	0.986	−0.018	0.758	−0.029	0.327	−0.031	
0–450	0.955	−0.021	0.598	−0.030	0.360	−0.030	
0–500	0.938	−0.023	0.810	−0.029	0.494	−0.029	
0–550	0.910	−0.024	0.707	−0.030	0.512	−0.029	
0–600	0.897	−0.026	0.650	−0.030	0.194	−0.031	
0–650	0.876	−0.026	0.266	−0.031	0.218	−0.030	
0–700	0.823	−0.027	0.314	−0.031	0.282	−0.030	
Notes.

Fij coefficient of kinship

P probability of autocorrelation at stand level

CI confidence interval

+ significant results after Bonferroni correction

# significant autocorrelation detected in the 50–75 m distance class in the PC-MA stand

## significant autocorrelation detected in the 50–100 m distance class in the PC-MA stand

CE index of aggregation

Asterisks indicate significant differences: ** at the 99.0% and *** at 99.9% level ns, not significant.

Table 4 Analysis of the genetic spatial structure within the Pinus cembroides seed stands PC-CT and PC-AD, considering the first class in distance class widths from 25–700 m, with 999 permutations and a confidence interval of 99%.

The analysis was conducted using SPAGeDi v1.4 (Vekemans & Hardy, 2004).

	Fijvs. spatial distance	
Distance	PC-CT	PC-AD	
	CE= 1.05 ns	CE= 0.31***	
	P(Fij) <CI	Fij	P(Fij) <CI	Fij	
0–25	0.903	0.072	0.964	0.010	
0–50	0.981	0.053	0.953	0.010	
0–75	0.464	−0.031	0.966	−0.003	
0–100	0.127	−0.041	0.999	0.005	
0–125	0.394	−0.031	0.996	−0.004	
0–150	0.329	−0.032	0.983	−0.012	
0–175	0.345	−0.032	0.997	−0.011	
0–200	0.260	−0.033	0.997	−0.013	
0–225	0.157	−0.034	0.998	−0.013	
0–250	0.357	−0.031	0.995	−0.016	
0–275	0.308	−0.031	0.997	−0.016	
0–300	0.090	−0.033	0.997	−0.016	
0–350	0.065	−0.034	0.995	−0.017	
0–400	0.347	−0.030	0.999	−0.016	
0–450	0.603	−0.029	0.996	−0.036	
0–500	0.734	−0.028	0.998	−0.019	
0–550	0.805	−0.028	0.998	−0.019	
0–600	0.829	−0.028	0.999	−0.020	
0–650	0.943	−0.028	0.998	−0.022	
0–700	0.868	−0.028	0.998	−0.023	
Notes.

Fij coefficient of kinship

P probability of autocorrelation at stand level

CI confidence interval

CE index of aggregation

Asterisks indicate significant differences: *** at 99.9% level; ns, not significant.

At the large scale (among seed stands), the individuals in the distance classes of 0–6 km, 0–7 km, 0–8 km and 0–86 km were genetically more closely related than random individuals from the “reference group” calculated by Fij. On the other hand, the individuals in other distance classes (0–168 km, 86–172 km, 172–258 km and 258–344 km) were found to be genetically significantly different, according to the Fij values (Table 5).

Table 5 Analysis of the genetic spatial structure among the Pinus cembroides seed stands, considering a distance class size of 86 km, 999 permutations and a confidence interval of 99%.

The analysis was conducted using SPAGeDi v1.4 (Vekemans & Hardy, 2004).

Fijvs. spatial distance (SPAGeDi v1.4)	
Fij	P(Fij) < CI	P(Fij) > CI	MT	
−0.000003	0.9999 (0–86)+	0.9999 (86–172)+
0.9999 (172–258)+
0.9999 (258-344)+	4,132	
Notes.

Fij coefficient of kinship

GD mean genetic distance

P(r) probability of autocorrelation per stand (in distance class, m)

CI confidence interval

MT minimal pairs of trees for class (class distance)

+ significant results after Bonferroni correction

At the large or landscape level (among seed stands, 6 to 168 km distances), the Principal Coordinate Analysis (PCoA - Coord. 1 vs. 2) clearly separated the five stands into four clusters (Fig. 2): PC-CT along with PC-MM located in Durango, PC-MA (Chihuahua), PC-AD (Durango) and PC-BQ (Chihuahua). However, the PCoA with Coordinate 1 vs. 3 revealed only three groups, and the three seed stands from Durango were grouped together (Fig. 3). The first two coordinates explained 69.7% of the variation in AFLP, and the first three coordinates explained 93.4% of the corresponding variation.

Figure 2 Principal Coordinates Analysis (PCoA) (coordinate 1 vs. 2) showing the genetic separation between the five seed stands of Pinus cembroides: Location with abbreviation of seed stand name: Mesa Azul (PC-MA): grey diamond, Baquiriachi (PC-BQ): black square, Los Adobe (PC-AD): grey circle, Mesa de la Majada (PC-MM): white diamond and Cordón del Toro (PC-CT): grey triangle.

Figure 3 Principal Coordinates Analysis (PCoA) (coordinate 1 vs. 3) showing the genetic separation between the five Pinus cembroides seed stands: Mesa Azul (PC-MA): grey diamond, Baquiriachi (PC-BQ): black square, Los Adobe (PC-AD): grey circle, Mesa de la Majada (PC-MM): white diamond and Cordón del Toro (PC-CT): grey triangle.

In a PCoA at the individual level, the first two coordinates explained a much lower percentage of the variation in AFLP (4.4 and 3.1%). Together the first three coordinates explained 10.1% of the variation. By plotting the first two coordinates, the 173 individual PC trees were not generally divided in separate groups. PC-MM had the largest mean TD, PC-BQ had the largest mean GD and PC-MA had the smallest mean GD (Fig. 4).

Figure 4 Principal Coordinates Analysis (PCoA) (coordinate 1 vs. 3) showing the genetic separation between the individuals in the five Pinus cembroides seed stands: Mesa Azul (PC-MA): grey diamond, Baquiriachi (PC-BQ): black square, Los Adobe (PC-AD): grey circle, Mesa de la Majada (PC-MM): white diamond and Cordón del Toro (PC-CT): grey triangle.

Discussion

The results showed non-significant autocorrelation in 80% (four out of five) of the natural seed stands studied (Tables 3 and 4), i.e., a mainly random distribution in the space of the genetic variants of the Pinus cembroides at the population level. The results thus did not confirm our hypothesis of weak SGS at the population level. Other studies of several Mexican pine species such as P. cembroides, P. discolor, P. durangensis, P. teocote (Hernández-Velasco et al., 2017), P. arizonica and P. cooperi (Friedrich et al., 2018) also detected no too weak SGS at the local scale. Especially for P. cembroides, Hernández-Velasco et al. (2017) found no SGS and smaller mean Tanimoto distances (mean 0.52) in three small (≤12.1 ha), closely spaced (≤10 km) seed stands.

These results can be explained by the random patterns and large distances over which pollen and seeds are dispersed (Ennos, 1994), along with a breeding system with a low selfing rate, non-significant local genetic drift and selection, and low population density supporting larger gene dispersal distances (Vekemans & Hardy, 2004). This particularly applies to P. cembroides, a species with wind-dispersed pollen, animal-dispersed seed and generally low population density (Little Jr, 1977; McCune, 1988; Tomback & Linhart, 1990), all of which favour gene flow (Vekemans & Hardy, 2004). This explanation is supported by the almost always negative pairwise kinship coefficients (Fij) (Tables 3 and 4) and greater genetic diversity than in other Mexican pine species (Wehenkel et al., 2015); (Hernández-Velasco et al., 2017). Furthermore, the overlapping seed shadows and demographic mortality may also result in non-significant SGS (Hamrick, Murawski & Nason, 1993; Epperson & Alvarez-Buylla, 1997; Parker et al., 2001; Fuchs & Hamrick, 2010). In our study, the trees sampled in each stand were heath, adult, dominant and occurred in the older age group; it is possible that SGS would have been revealed if other (younger) age classes had been sampled, as with Cecropia obtusifolia (Epperson & Alvarez-Buylla, 1997). Moreover, the sampling strategy may also be a decisive factor (see below).

Significant spatial autocorrelation along with almost always positive Fij was only observed in the first distance classes in the 0–100 to 0–200 m class widths. The autocorrelation was also almost significant (together with a positive Fij) in the 0–75 m distance class, indicating family structure (in these distance classes). However, the first classes in the smallest widths tested (0–25 and 0–50 m) showed no significant SGS along with the two most negative Fij in this study (Table 3). However, the lack of significance may have been due to an insufficient number of tree pairs (i.e., insufficient repetitions) in these small class sizes. We therefore assumed that there was no significant family structure in smaller groups of adult, dominant and older P. cembroides in PC-MA. However, in the first classes in larger distance class widths (i.e., 0–225 to 0–700 m) in PC-MA, also with sufficient tree pairs for statistical analysis, this index became more and more negative (Table 3). Thus, the P. cembroides tree pairs became more and more genetically different and larger from a distance of 200 m, indicating weakened gene flow, probably caused by isolation by distance gene dispersal.

The significant SGS, detected only in PC-MA, may have been caused by the range of spatial scales and the sampling strategy in this stand. In contrast to the other four stands, PC-MA was linear and had the longest linear expansion (2,300 m rather than <1,300 m) (Fig. 5), i.e., the range of spatial scale encompassed by the sample was the largest in this study. In addition, the larger the range, the better the regression performance in the SGS analysis (Heuertz et al., 2003; Vekemans & Hardy, 2004). Moreover, the sampling strategy was similar to a line-transect scheme, which may perform slightly better than the simple-random scheme used in our study (Zeng et al., 2010). The significant pattern of SGS in PC-MA is equal to that observed in the vast majority of studies which have often detected SGS only at the smallest spatial scales studied. This finding has usually been interpreted as a consequence of an isolation-by-distance process with restricted seed dispersal within plant populations (e.g., Sokal & Wartenberg, 1983; Streiff et al., 1998; Sork et al., 2002; Vekemans & Hardy, 2004).

Figure 5 Map of the Pinus cembroides seed stand Mesa Azul (PC-MA) and the positions of the 34 sample trees genetically analyzed (black circles).

Source: ESRI Inc. (1999–2012). ArcGIS for Desktop 10. USDA Natural Resources Conservation Service.

Significant SGS was observed (Table 5) at the landscape scale (among the five seed stands), supporting the theory of isolation by distance as a consequence of restricted pollen and seed dispersal (Wright, 1938). However, the lack of outlier AFLP loci indicated that local selection was not an important factor in relation to SGS in this study (Epperson, 1992).

The biological meaning of the spatial genetic patterns can easily be misinterpreted without highly variable markers, appropriate sampling or detailed ecological data such as the age, sex and social status of the individuals sampled (Double et al., 2005). We used (i) stable markers, but with many loci (281 AFLP) (Vekemans & Hardy, 2004; Cavers et al., 2005), (ii) the social status (dominant trees), (iii) sex (monoecious) and (iv) proximate age (the higher age-group) of the individuals sampled. However, the sampling number and structure may represent two of the weakest points in this study. More than 35 samples per stand and a larger range of spatial scale are probably needed to detect weak SGS, which is expected in tree species with wind-dispersed pollen and animal-dispersed seed (Hamrick, Murawski & Nason, 1993; Vekemans & Hardy, 2004), such as the wingless seed of P. cembroides (Richardson, 2000). The line-transect scheme may also be the best sampling strategy (Zeng et al., 2010).

Although we detected SGS at landscape (among seed stands) level, our study is not strong enough in relation to coverage of the study site and intermediate distance intervals, as only five stands were studied. In addition, the range of spatial scales included in the sample was not maximized (Vekemans & Hardy, 2004). Moreover, if genetic variation is arranged clinally, then sampling only at well-separated stands in this continuum may create an artificial set of “isolated populations” that nevertheless are connected by the unsampled trees in between. Furthermore, if the distance classes are set exactly to the distances that separate the sampling points, this effect may be exaggerated.

Conclusions

On the basis of the SGS analysis at the local scale and also assuming that SGS was weak, we concluded that seed stands of P. cembroides may represent small-scale units. However, a minimum width of 225 m is required in these stands to produce seeds without significant loss of genetic variation, because all classes with distance class size 225 m or larger did not show significant SGS and Fij was negative. Indeed, the Mexican norm for forest germplasm (NMX-AA-SC-169-2016) requires a minimum stand size of two hectares (e.g., 141 × 141 m), which, however, may be too small.

On the basis of the SGS analysis, we conclude that establishment of a close network of PC seed stands separated by a maximum distance of 6 km can prevent greater loss of local genetic variants and alteration of SGS. However, we estimate that establishing such a high density of seed stands would not be economically feasible in forest management and conservation plans.

In addition, all of the AFLP loci identified are probably neutral markers. They may consequently be more variable and more differentiated than adaptive markers, as a result of isolation and genetic drift (Petit et al., 2001; McKay & Latta, 2002). Definition of seed collection zones by differences in genetic adaptation could therefore result in an overly dense network of seed stands.

Supplemental Information

Supplemental Information 1 Raw data, including the AFLP and coordinates (UTM, R13) of the five Pinus cembroides populations studied

Click here for additional data file.

Additional Information and Declarations

Competing Interests

Author Contributions

Data Availability

Christian Wehenkel is an Academic Editor for PeerJ.

Luis C. García-Zubia performed the experiments, analyzed the data, prepared figures and/or tables, approved the final draft.

Javier Hernández-Velasco performed the experiments, analyzed the data, prepared figures and/or tables, approved the final draft.

José C. Hernández-Díaz analyzed the data, prepared figures and/or tables, authored or reviewed drafts of the paper, approved the final draft.

Sergio L. Simental-Rodríguez performed the experiments, prepared figures and/or tables, approved the final draft.

Carlos A. López-Sánchez analyzed the data, prepared figures and/or tables, authored or reviewed drafts of the paper, approved the final draft.

Carmen Z. Quiñones-Pérez analyzed the data, authored or reviewed drafts of the paper, approved the final draft.

Artemio Carrillo-Parra analyzed the data, prepared figures and/or tables, approved the final draft.

Christian Wehenkel conceived and designed the experiments, analyzed the data, contributed reagents/materials/analysis tools, authored or reviewed drafts of the paper, approved the final draft.

The following information was supplied regarding data availability:

Raw data are available in the Supplemental File.

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
