# Peer review of "Spatial genetic structure in Pinus cembroides Zucc. at population and landscape levels in central and northern Mexico"

_PeerJ, doi:10.7717/peerj.8002_

## Round 0.1 · original submission · Major Revisions

Both reviewers raise questions on your spatial genetic structure (SGS) analysis. Addressing this issue will be critical for the publication of your manuscript.

Reviewer 1 ·

Basic reporting

Dear Ph.D. Robert Winkler
Associate Editor

This study investigated the spatial genetic structure (SGS) in fine scale (within populations) and large scale (landscape) of five Pinus cembroides populations, using the dominant AFLP gene marker, aiming to establish guidelines for conservation and management of the species. The sample size ranged from 34 to 45 trees per population. The SGS analysis was carried out, using three different relatedness estimators between pairwise individuals and four different softwares. The results suggest low or no significant SGS within stands, but the pattern of gene dispersal of isolation by distance among stands.

The study is original, the results are discussed and paper fits the PeerJ scope. However, I have some suggestions and corrections to improve the paper.
Best regards,
The reviewer

Experimental design

The weak SGS in fine scale may an artifact of the low sample within the stands, as pointed also by authors, as well as due the fact that the study is based in samples of old individuals. To robust analysis based on 100 AFLP loci, samples of at least 150 individuals are recommended. The study is based on 283 loci, as recognize by authors, a higher sample size is necessary for SGS analysis (line 295), probably about 60 individuals. Old individuals may not be related due to many stochastic (random mortality, diseases, predation etc) and determinist factors (inbreeding depression, natural selection), decreasing SGS. However, SGS may occur in individuals of the younger classes. Thus, the results may only be valid for old trees. Alternatively, to overcome the problem of low sample size within populations, the intrapopulational SGS can be analyzed joint all individual and using shorter distance classes (for example 50, 75, 100, 150, 200, etc up to 700 m). Furthermore, more information about the populations is needed to understand the study, such as population density, origin of the population (natural or artificial and originate from plantation, probable age of the sampled trees).

Other point to be considered. Why did you use different methods to investigate the SGS in fine scale? Please, justify. I did not find any advantage in this strategy to analyses the data. I suggest use only Fij, implemented in Spagedi and to estimate the Sp-statistic for compare with other studies.

Validity of the findings

no comment

Additional comments

Manuscript ID: Peerj 39338-v0

Title: Spatial genetic structure in Pinus cembroides Zucc. at population and landscape levels in central and northern México

Dear Ph.D. Robert Winkler
Associate Editor

This study investigated the spatial genetic structure (SGS) in fine scale (within populations) and large scale (landscape) of five Pinus cembroides populations, using the dominant AFLP gene marker, aiming to establish guidelines for conservation and management of the species. The sample size ranged from 34 to 45 trees per population. The SGS analysis was carried out, using three different relatedness estimators between pairwise individuals and four different softwares. The results suggest low or no significant SGS within stands, but the pattern of gene dispersal of isolation by distance among stands.

The study is original, the results are discussed and paper fits the PeerJ scope. However, I have some suggestions and corrections to improve the paper. The weak SGS in fine scale may an artifact of the low sample within the stands, as pointed also by authors, as well as due the fact that the study is based in samples of old individuals. To robust analysis based on 100 AFLP loci, samples of at least 150 individuals are recommended. The study is based on 283 loci, as recognize by authors, a higher sample size is necessary for SGS analysis (line 295), probably about 60 individuals. Old individuals may not be related due to many stochastic (random mortality, diseases, predation etc) and determinist factors (inbreeding depression, natural selection), decreasing SGS. However, SGS may occur in individuals of the younger classes. Thus, the results may only be valid for old trees. Alternatively, to overcome the problem of low sample size within populations, the intrapopulational SGS can be analyzed joint all individual and using shorter distance classes (for example 50, 75, 100, 150, 200, etc up to 700 m). Furthermore, more information about the populations is needed to understand the study, such as population density, origin of the population (natural or artificial and originate from plantation, probable age of the sampled trees).

Other point to be considered. Why did you use different methods to investigate the SGS in fine scale? Please, justify. I did not find any advantage in this strategy to analyses the data. I suggest use only Fij, implemented in Spagedi and to estimate the Sp-statistic for compare with other studies.

Best regards,
The reviewer

Minor comments
Line 55. Where you read … isolation by distance …. I suggest ….. isolation by distance gene dispersal …

Lines 56-57. This sentence needs a citation.

Line 77. Where you read … thus providing few loci such as allozyme …. I suggest ….. thus, gene markers such as allozymes, which provide few loci are not adequate.

Line 105. Where you read … isolation by distance among…. I suggest ….. isolation by distance gene dispersal among…

Lines 107 and 118. Where you read … Pinus cembroides …. I suggest ….. P. cembroides …

Line 113. Please, delete “Add your introduction here.”

Lines 118-125. More information about populations is needed to understand the study, such as population density. In addition, Wehenkel et al. (2015) is written in German and not everyone can read to understand their study.

Line 222. The species name must be in italic (P. cembroides).

Line 244. Where you read … (4.4 % and 3.1 %) …. I suggest ….. (4.4 and 3.1%) …

Line 308. I suggest changing the term conclusions to conservation guidelines because the results continue to be discussed.

Annotated reviews are not available for download in order to protect the identity of reviewers who chose to remain anonymous.

Reviewer 2 ·

Basic reporting

Overall accepted text clarity and structure.

Experimental design

Acceptable two-levels of sampling scales. OK DNA marker techniques.

Validity of the findings

Findings supported by data.

Additional comments

Main Coomments:

(1) Discussion is needed to be expanded about possible explanations of the lack of small-scale SGS. Perhaps be more explicit about your possible explanation. Say perhaps that the fact that trees sampled were only on “trees in each stand which were adult, dominant and occurred in the older age group”. Perhaps if other (younger) age classes were sampled, SGS could be revealed, at it happened for Cecropia obtusifolia (Epperson & Alvarez‐Buylla, 1997)

(2) About conclusions: Are you sure that is worth to be engaged in a network of so close seed stands, distant less to 6 km, and use the seeds in a radius of only 3 km? Is it not exaggeratedly narrow and restrictive managing strategy, especially when, as you say, based on adaptive genetic variation, scale of differences likely would be much larger? Your forest management/conservation strategy seems to me not justified and if put in place (which I doubt), too expensive with dubious positive cost/benefit balance. Consider more about the reality of forest management.

(3) Conclusions are too long. Specially the first paragraph: too much references and too few facts based on your results. Make something more synthetic, describing your more important findings, based directly on your results.

Specific comments.
45-46. What do you mean with?: “these stands should have a minimum width of at least 110 m to product seed” Do you mean to produce seeds?

56-67. Reference needed to support that.

72. Explain briefly wjay you mean with “a mixed sampling strategy”. Sampling at different intensities, e.g. sampling plots at small and a large scale, such as, say, every 50 m combined with plots every 50 km??

96 . Luna-cavazos -> Luna-Cavazos.

113. “Add your introduction here.” (¿???????????)

222. Pinus cembroidesin italics.

237. No so clear what you mean with “At the seed stand level”. Perhaps: “at the large-scale -landscape- level (among seed stands, 6 to 168 km distance)..”

255. “..also detected no to weak SGS..” (?) Or “..also detected no too weak SGS..”?

267-270. About the lack of small-scale SGS. Perhaps be more explicit about your possible explanation. Say perhaps that the fact that trees sampled were only on “trees in each stand which were adult, dominant and occurred in the older age group”. Perhaps if other (younger) age classes were sampled, SGS could be revealed, at it happened for Cecropia obtusifolia (Epperson & Alvarez‐Buylla, 1997).

319. “to product seed” ¿? Or “to produce seeds”

325 “a maximum distance of 6 km” or you meant “a minimum distance of 6 km” ??

325-326. Are you sure that is worth to be engaged in a network of so close seed stands, distant less to 6 km, and use the seeds in a radius of only 3 km? Is it not exaggeratedly narrow and restrictive managing strategy, specially when, as you say, based on adaptive genetic variation, scale of differences likely would be much larger? Your forest management/conservation strategy seems to me not justified and if put in place (which I doubt), too expensive with dubious positive cost/benefit balance. Consider more about the reality of forest management.

Figure 1. Panel of Mexico has too low resolution.

---

## Round 0.2 · accepted · Accept

Both reviewers agree that you replied adequately to all their comments

Reviewer 1 ·

Basic reporting

I quickly reviewed the new version of the manuscript and realized that all my suggestions were accepted. Thus, in my opinion, the manuscript can be published in the present form.

Experimental design

Nothing to comment.

Validity of the findings

Nothing to comment.

Additional comments

Nothing to comment.

Reviewer 2 ·

Basic reporting

Overall acceptable manuscript

Experimental design

ALthough (as allways) a more intensive sampling of younger trees would be desirable, with the sampling that they did, results are supported and sicussion apply well for adult/older trees, as pointed out by the other reviewer. Thus, it is acceptable.

Validity of the findings

Results and conclusions supported, although I do not fully agree with so much restrictions of seed movement.

Additional comments

Auhtors responded in an acceptable way to my own comments and seems to me the responses to the other reviewer are reasonable.
Paper can be accepte in its present form.